


**The vulnerability of buildings to a large-scale debris flow and outburst**
**flood hazard chain that occurred on 30 August 2020 in Ganluo,**
**Southwest China**
Li Wei [1], Kaiheng Hu[1], Shuang Liu[1], Nan Ning [1,2], Xiaopeng Zhang[1,2], Qiyuan Zhang[1,2], Md Abdur Rahim[1,2,3]
[1] Key Laboratory of Mountain Hazards and Earth Surface Processes, Institute of Mountain Hazards and Environment, Chinese
Academy of Sciences,Chengdu 610041,China
[2] University of Chinese Academy of Sciences, Beijing 100149, China.
[3] Department of Disaster Resilience and Engineering, Patuakhali Science and Technology University, Dumki, Patuakhali-8602,
Bangladesh
*Correspondence to*: Kaiheng Hu (khhu@imde.ac.cn)
**Abstract:** In mountainous areas, damage caused by debris flows is often aggravated by
subsequent dam-burst floods within the main river confluence zone. On 30 August 2020, a
catastrophic disaster chain occurred at the confluence of the Heixiluo Gully and Niri River in
Ganluo County, Southwest China, that consisted of a debris flow, the formation of a barrier lake
and subsequent dam breach that flooded the community. This study provides a comprehensive
analysis of the damage to buildings resulting from the sequential occurrence of debris flow and
dam-burst flood. The peak discharge of the debris flow in the gully mouth reached 1937 m³/s, and
the change in the main river channel resulting from the dam-burst flood, which had a peak
discharge of 2273 m³/s, resulted in a fourfold increase in the extent of flood inundation compared
to an ordinary flood. Three hazard zones were established based on the building damage patterns:
(I) primary debris flow burial; (II) secondary dam-burst flood inundation and (III) sequential
debris flow burial and dam-burst inundation. Vulnerability curves were developed for Zone (II)
and Zone (III) using impact pressures and inundation depths, and a vulnerability assessment chart
is presented that contains the three damage categories. This research addresses a gap in the
vulnerability assessments of debris flow hazard chains and can support in future disaster
mitigation within confluence areas.
**Keywords**: Multi-hazard risk, Debris flow, Dam-burst flood, Building damage, Vulnerability
analysis.

## 1 Introduction

In mountainous areas, debris flows frequently block rivers and form temporary dammed lakes.
The subsequent breach of these dammed lakes can result in a vast flash flood (Yan et al., 2020).
The hazard chain consisting of debris flows and subsequent dam-burst floods usually devastates
residential buildings in confluence zones. For instance, a large-scale debris flow occurred in the
Wenjia Gully in Sichuan Province, Southwest China, on 13 August 2010 and completely blocked


the Mianyuan River, which formed a dammed lake 1650 m long, 420 m wide, and 12 m deep.
Then, the dammed lake breached and caused 7 fatalities and extensive damage to 479 houses (Yu
et al., 2013).

Multi-hazard analyses that incorporate potential hazard interactions has gained significant

attention in recent years (Liu et al., 2015; Gallina et al., 2016; Tilloy et al., 2019; Luo et al., 2023).
However, the vulnerability assessments in risk analysis rarely consider the effects of hazard
interactions (Luo et al., 2023). Argyroudis et al. (2019) introduced a new methodology for
evaluating the vulnerability of transport infrastructure to multiple hazards. This approach is
comprised of six steps and includes numerical and fragility models. Progress has been made in
assessing the risk of buildings exposed to multiple hazards by considering the interaction between
an earthquake and other hazards, such as dam breaks, flash floods, and tsunamis. Korswagen et al.
(2019) proposed a methodology for assessing structural damage resulting from coupled hazards
and used it to assess the vulnerability of a masonry building subjected to an earthquake and an
earthquake-triggered dam break. Furthermore, Park et al. (2012) developed collapse fragility
curves for earthquake and tsunami effects using a numerical model. Gautama and Dong (2018)
outlined the vulnerability of vernacular stone masonry buildings to the flash floods that occurred
after the Gorkha earthquake. Residential buildings in Nepal were found to have up to 300%
damage resulting from the combined earthquake and subsequent flash flood. Petrone et al. (2020)
simulated the response of reinforced concrete frames to earthquake and tsunami inundation,
yielding fragility curves that showed a median decrease of less than 15% in terms of tsunami
resistance when exposed to cascading hazards as compared to tsunami-only fragility functions.

The evaluation and mitigation of the multiple risks posed by debris flows and dam-burst

floods in a confluence zone require a multi-risk analysis that considers hazard interactions and
their cumulative effects on building vulnerability. Most studies on debris flow and dam-burst
floods mainly focus on numerical simulations and the evolving processes of hazard chains (Nin et
al., 2022; Chen et al., 2022), but studies on the vulnerability of building to hazard chains are
scarce. The vulnerability of buildings to the cumulative impact of debris flow and flash flood may
differ from the sum or sequence of vulnerability resulting from a single debris flow or flash flood
(Kappes et al., 2012). The effect that simultaneous hazards have on building vulnerability remains
inadequately addressed, with only a few studies available (Kappes et al., 2012). Luo et al. (2020)
proposed a framework for developing physics-based vulnerability models for buildings exposed
to multiple surges of debris flows. Cumulative damage effects resulting from sequentially
occurring debris flows were quantified by assessing the physical damage from primary debris





flows. However, this approach may not apply directly to the debris flow-dam-burst flood hazard chain.

Field investigations have shown that the pattern of damage to buildings in the confluence area of debris flow and flood are not consistent with those from the debris fan or on the floodplain. Debris flow usually causes devastating damage to settlements on the fan, and the subsequent dam-burst flood significantly increases the damage (Xu et al., 2014; Yu et al., 2013). The risk amplification and cumulative effect on building vulnerability resulting from successive debris flows and dam-burst floods are not entirely clear. Therefore, in-depth analysis is essential for assessing the risks posed by the debris flow hazard chain in order to develop a successful emergency management plan.

On August 30, 2020, a catastrophic debris flow and dam-burst flood occurred in the Niri River, Ganluo County, Sichuan Province, Southwest China. This study aims to comprehensively analyze the damage to buildings caused by the Heixiluo debris flow-dam-burst flood disaster chain. Firstly, we calculated the dynamic characteristics of the debris flow and outbreak flood damage. We then systematically investigated and summarized the building damage characteristics and analyzed and compared the vulnerability of buildings considering different damage patterns. Finally, we discuss how the damage was amplified by the chain and offer suggestions for hazard mitigation.

## 2 Study area

The study area is located in Ganluo County, Sichuan Province, Southwest China, which includes the Heixiluo Gully and the confluence area along the Niri River. Ganluo County lies north of the Liangshan Yi Autonomous Prefecture, occupying the alpine canyon zone in the transitional region between the western margin of the Sichuan Basin and the Qinghai-Tibet Plateau (Fig. 1). The geographic boundaries of the study area span from 102°27' to 103°01' east longitude and from 28°38' to 29°18' north latitude. Ganluo County covers a total area of 2150.97 km² and had a permanent population of 205,991 at the end of 2020.



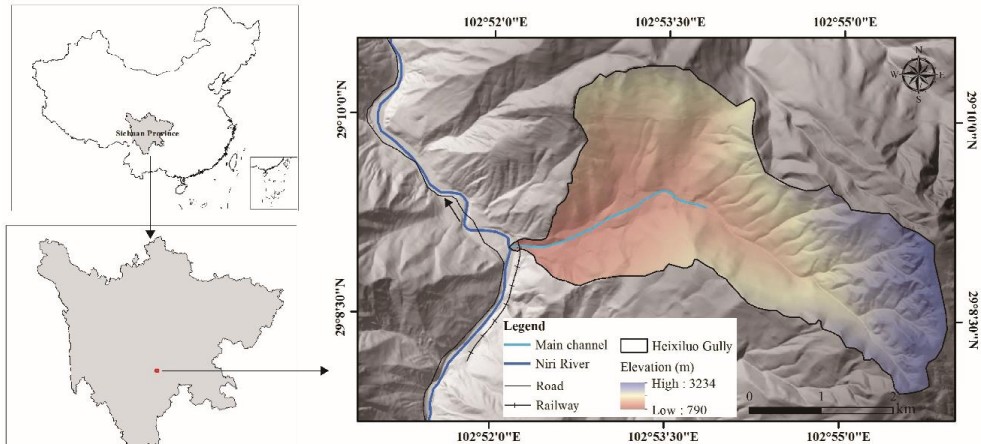

**Figure 1** Location of the study area including the Heixiluo Gully and Niri River.

Ganluo County consists of an erosional tectonic landform that is defined by two primary structures, namely Sichuan-Yunnan north–south structure and the Qinghai-Tibet Yunnan zeta-type structure. The mountain and river systems flow from south to north due to the folds, uplift, and fractures of the Hengduan Mountains and the strong disruptive effect that widely distributed rivers, undulating hills, ravines, and cliffs have on the study area. The valleys, which are characterized by a V-shaped cross-section, have considerable depths that typically exceed 1000 meters.

Folds are ubiquitous in the study area, and the N–S trending Teke fault, Suxiong anticline, and Maanshan anticline are excellent examples of these typical geological structures. These faults were last active during the early and middle Pleistocene and there is no discernible evidence that they were active during the late Quaternary period. The exposed strata in the study area are primarily comprised of Quaternary strata (Q), Presinian Ebian Group (Pteb), and Lower Sinian Suxiong Group (Zas). The upstream area is mainly occupied by sandstone, whereas rhyolite and tuff dominate the main part of the catchment, with slate occupying the left downstream area. The study area is situated in a seismically active region. The peak ground acceleration in the study area is 0.15 g, and the peak period of the seismic response spectrum is 0.45 s. Between 1327 and 1975, nearly 147 Ms ≥ 2.5 earthquakes were recorded, including 15 Ms ≥ 5.0 earthquakes with the highest magnitude of 7.5.

The Niri River is a first-order tributary of the middle reaches of the Dadu River and flows from south to north and over an elevation range of 1800-2200 for most of the areas. The highest elevation in the river basin is 4700 m, and the lowest elevation is 1170 m. The study area has a subtropical monsoon climate. Summer is humid and hot with abundant rainfall, while winter and

spring are warm and dry. The precipitation is distributed unevenly throughout the year. The
rainfall is concentrated from April to October, with an average rainfall of 901.9 mm, accounting
for 93.14% of the average annual rainfall. The precipitation varies significantly with elevation,
with an annual precipitation of 968 mm, and the maximum hourly rainfall and ten-minute rainfall
recorded are 40.3 mm and 14.8 mm, respectively.

## 3 Hazard chain

The Heixiluo Gully is located on the right bank of the Niri River in Suxiong town, Ganluo
County (Fig. 1). The coordinates of the gully mouth are 29°09′47″N and 102°52′53″E and the
gully extends from the east to the west. The gully covers an area of 13.36 km$^2$ and is situated at a
moderate elevation on the mountainous landform. The catchment elevation ranges from 3220 m
above sea level (a.s.l.) to 760 m a.s.l., with a relative height of 2460 m. The main channel of the
gully stretches for 6.93 km, with an average gradient of 355‰. The average slopes for areas
above and below 1990 m are approximately 600‰ and 256‰, respectively.
The field investigation indicates that the area above elevation of 1990 m where the flash
floods provide hydrodynamic conditions are sufficient for forming debris flows. The gradient of
the valleys in this area is steep, with an average value of 600 ‰. The transportation zone is
mainly located between 820 m and 1990 m in elevation and occupies an area of 5.96 km$^2$. The
length of the main gully is 4.65 km, and the average gradient of the main gully is 252‰. Two
platforms were distributed at altitudes of 1160 m and 1030 m and divided the main channel of the
transportation zone into three parts. A narrow channel developed between the platform and the
deposition fan at 10230 m. The length and gradient of the channel are approximately 670 m and
243‰, respectively.
The debris flow was triggered by a once in a century short-term heavy rainfall event.
According to the precipitation data from two automated stations located 10 km away, the 24-hour
cumulative rainfall from 8:00 on 30 August was approximately 82.8 mm. The rainfall data
extracted from the Global Precipitation Measurement (GPM) rainfall product in the Heixiluo
Gully showed that the rainfall started on 29 August at 22:00 and lasted until 6:00 on 31 August
and delivered a total of 147.2 mm of rain. The hourly rainfall increased to 5.18 mm at 19:30 on
30 August, which triggered the debris flow due to the approximately accumulated 61.4 mm of
rainfall. The debris flow lasted approximately 40 minutes, and the rainfall intensity reached 6.63
mm/h (Fig. 2).



Heavy rainfall caused flooding in the Yanrun Hydrometric station (located 15 km upstream
from the study area), resulting in a peak discharge of 893 m$^3$/s (He et al., 2020), which was nearly
nine times of the average discharge.
The debris flows transported approximately 1,050,000 m$^3$ of material to the Niri River, forming
a temporary debris dam that breached after approximately 30 minutes, resulting in a massive flash
flood. The debris flow-flash flood event caused significant damages, including the destruction of
89 buildings, the Chengdu-Kunming railway bridge near the gully mouth, 1.2 km along national
road G245, and 5 highway bridges along the main river (Fig. 3).

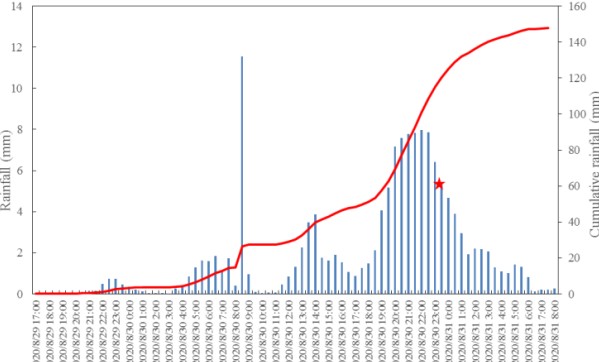


**Figure 2**  Hourly and cumulative rainfall on 29, 30, and 31 August 2020 extracted from the Global

159             Precipitation Measurement (GPM) rainfall product.

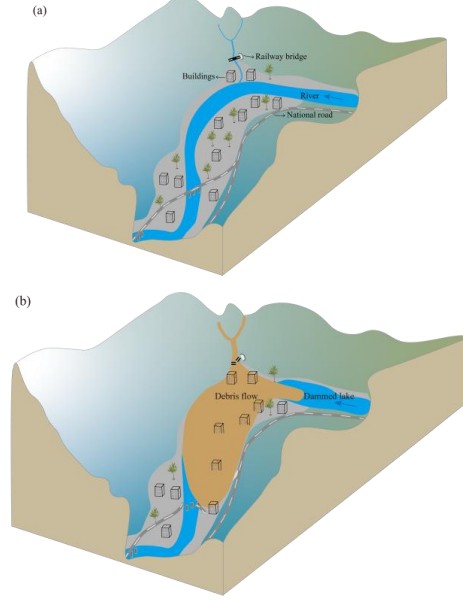

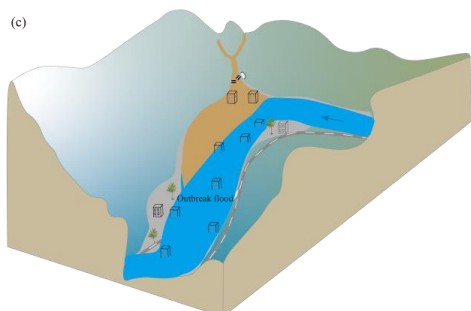

**Figure 3** Illustration of the multi-hazard chain: (a) river flow before the occurrence of debris flow; (b) debris flow blocks the river and forms a dammed lake that destroys the railway, roads and buildings; (c) the bursting of the dammed lake and buildings and road were damaged or inundated by the flood.

# 4 Data and methods

We conducted field investigations on the debris flow-flash floods that occurred on 31 August and 3 December 2020. The field survey mainly focused on the main transportation and deposition zones. Interviews, measurements, and aerial photography were conducted to investigate the formation and disaster mechanisms. The geomorphic settings of the Heixiluo Gully and adjacent Niri River were carefully measured and analyzed, including the channel width, deposition and erosion height, channel slope, and particle size distributions. The damage to buildings was also investigated by comparing the drone photos taken before and after the disaster.

## 4.1 Data collection

The Digital Elevation Modes (DEMs) collected before and after the event were used for hazard chain analysis. The pre-event DEM was converted from a 1:10000 topographic contour map provided by the Sichuan Bureau of Surveying, Mapping, and Geoinformation which had a spatial resolution of 10 m. The post-event DEM of the study area was produced by synthesizing high-resolution aerial images captured by a Dajiang unmanned aerial vehicle (UAV) on 3 December 2020. To calibrate the post-event terrain, 10 image control points that were not affected by the disaster were selected, and their elevation values were sampled from the pre-DEM and assigned as input conditions. The mean RMS error of georeferencing of the post-event DEM was within the usable range with a value of 0.1 m.

## 4.2 Methodology

The dynamic parameters of the debris flow and discharge of the dam-burst flood were calculated by the formulas presented in Table 1.




**Table 1** Models used in parameter calculation for this study

| Category of Calculation | Applied formula | Description parameters |
|---|---|---|
| Debris flow density (Hu et al., 2019) | $\gamma_c = -1320x^7 - 513x^6 + 891x^5 - 55x^4 + 34.6x^3 - 67x^2 + 12.5x + 1.55$ | x is the clay content in the debris flow sample. The average clay content in particles less than 0.005 mm in size accounts for 2.55%. |
| Debris flow peak discharge and velocity (Kang, 1987; Yang, 1985) | $Q = \frac{1}{n_c}AR^{\frac{2}{3}}J^{\frac{1}{2}}$ $n_c = \frac{1}{18.5H^{-0.42}}$ $U = \frac{Q}{A}$ | A is the cross-sectional area, R is the hydraulic radius, J is the channel bed gradient, and $n_c$ is the roughness coefficient for viscous debris flow. The method for calculating $n_c$ was deduced from analysis of viscous debris flows in Huoshao gully in China. |
| Dam-burst flood discharge | $Q = \frac{1}{n}AR^{\frac{2}{3}}J^{\frac{1}{2}}$ | A is the cross-sectional area, Rn is the hydraulic radius, J is the channel bed gradient, and n is the Manning roughness coefficient. The values of A, $R$, and J were directly measured by the field investigation. |

Dam-burst flood hydraulics were simulated by HEC-RAS 5.0.7 (Hydrologic Engineering
Center,2016) using the post-event DEM. The computation procedure employed a one-
dimensional steady flow simulation and assumed a subcritical flow regime. The boundary
conditions are established at all the ends of the river nodes by entering the normal depth value.
The initial conditions were set using the corresponding discharge of the dam-burst flood
estimated at a typical river section using Manning's equation. Manning's *n* coefficient, expansion,
and contraction coefficients account for flow energy losses in HEC-RAS. The expansion and
contraction coefficients were specified as 0.1 and 0.3, respectively, based on the width of the
channel reaches. Considering the debris flow deposition in the main river, Manning's *n* values of
the river channel and floodplain were assigned the same value of 0.05. This value is the suggested
value for main channels with clean, winding, pools, shoals, stones, ineffective slopes, and
sections.
The dam-breaching peak flow rate was approximately 3.1 times the upstream inflow flood,
with a discharge of 893  m³/s. Therefore, the input discharge for the ordinary flood and dam-burst
flood in the HEC-RAS model were set as 893 m³/s and 2273 m³/s, respectively. Due to the
difficulty of acquiring terrain data for the initial stage of the dam breach, it was assumed that the
peak discharge of the dam-burst flood formed the post-event terrain, which was adopted to
simulate the dam-burst flood. To analyze the impact of debris flows on river dynamics, we also
simulated an ordinary flood unaffected by debris flows using the pre-event DEM. The input
discharge for this simulation was 893 m³/s, which was recorded at upstream hydrological
observation stations located approximately 15 km from Heixiluo Gully. The Manning's *n* values
for the river channel and floodplain were 0.4 and 0.2, respectively. These values are the suggested
values for main channels that are clean and winding, have some pools and shoals, some weeds



and stones, and have flood plains for cultivated areas but are free of crops. The data applied to the
flood calculations are presented in Table 2.
**Table 2** Data used in the flood simulation

| Floodprocessing | Data | Data source | Manning's n value | Expansion and contraction coefficients |
|---|---|---|---|---|
| Debris flow dam-burst flood | Topography | Post-event DEM of the river channel | 0.4 (river channel), 0.2 (floodplain) | 0.1 (expansion coefficient) 0.3 (contraction coefficient) |
| | Discharge | Estimated by Manning's equation in a typical section | | |
| Flood not affected by debris flow | Topography | Pre-event DEM of the river channel | 0.5 (river channel and floodplain) | |
| | Discharge | Record in the Yanrun Hydrometric station (located upstream 23 km from Heixiluo Gully) | | |

A vulnerability curve was developed to describe the relationship between the hazard intensity
and the degree of damage to the buildings. Following the classification of the damage degrees
proposed by Hu et al. (2012), the degree of damage to buildings caused by multi-hazards was
determined through a comprehensive analysis of photographs taken on site and aerial images
collected over the disaster scene. Hazard intensity parameters were applied, such as flow depth
and impact pressure, with impact pressure calculated as $P = \rho v^2 + 0.5\rho gh$, where P is the
impact pressure, $\rho$ is the flow density, $v$ is the velocity, and $h$ is the flow depth. The deposition
depth of the debris flow was obtained by field investigation, while the velocity was calculated
using the method outlined in **Table 1**. The maximum flow depth and velocity of the flood were
extracted from the HEC-RAS model. A nonlinear regression analysis was conducted using a
logarithmic form expression to relate the vulnerability to the intensity parameters of the hazard.
## 5 Results
**5.1 Dynamic characteristics of the debris flow and outbreak flood**
Samples of debris particles smaller than 10 cm were taken from three locations (see Fig. 4).
The particle size distribution of the debris flow samples is presented in Fig. 5. The calculated bulk
density of the debris flow is 1.825 g/cm³, which indicates a viscous debris flow (Kang et al.,

228 2004).




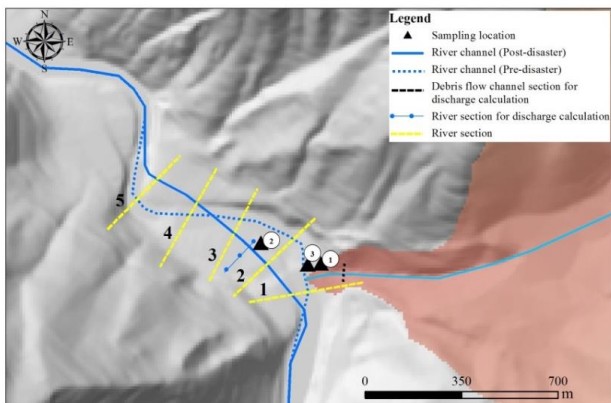


**Figure 4** Distribution of river and debris flow channel sections and debris flow sampling locations.

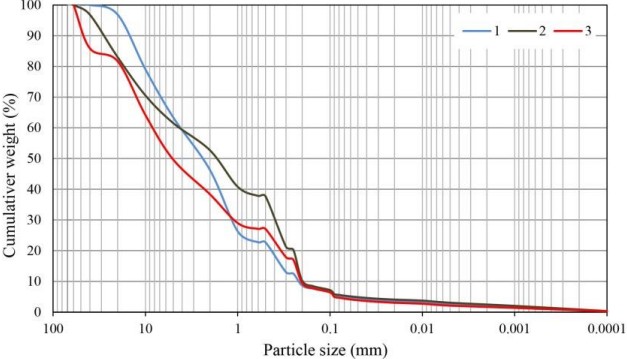


**Figure 5** Particle size distribution of debris flow samples.
The debris flow destroyed the Chengdu-Kunming railway bridge situated at the gully mouth
and had a flow depth of approximately 5 m and a section area of approximately 150 m$^2$. The
estimated peak discharge at the gully mouth using Kang's equation (1987) was 1611 m$^3$/s, which
resulted in a high impact pressure of 255 kPa.
The debris flow lasted for approximately 40 minutes and transported approximately
1,050,000 m$^3$ of sediment downstream. The deposition zone extended from the gully mouth to the
floodplain of the Niri River, covering a length of 320 m (Fig. 3). The area measured from the
UAV image was approximately 0.15 km$^2$. The thickness of the sediment deposits ranged from 5
m to 15 m, with an average value of 7 m. The debris flow flushed into the main river and blocked
the Niri River. The river channel was filled with sediment, which lead to the formation of a
dammed lake that raised the water level by 7-8 m. After 30 minutes, the unstable dammed lake
breached, which resulted in a massive flash flood.



The outburst of the debris flow lake caused a sharp increase in flood peak discharge. To
analyze the dynamic characteristics of the flood caused by the dam burst, we first used Manning's
hydraulic formula for open channel flow (presented in Table 1) to calculate the peak discharge.
Then, we selected empirical formulas for dam-burst floods to verify the discharge. A typical
section adjacent to buildings damaged by the flood was chosen for the calculation (Fig. 4). Based
on flood traces on the outer walls of buildings and the damaged height of buildings, the flood
depth was estimated to be 6 m. The cross-sectional area and hydraulic radius were calculated
according to the section geometry and channel profile. The channel bed gradient was determined
based on the longitudinal channel profile. The resulting peak discharge was 2737 m$^3$/s. Field
investigation revealed that the height of the debris flow dam was approximately 12 m. The
volume of the barrier lake was calculated based on the terrain data collected before the disaster.
The peak discharge was estimated using the empirical formula proposed by Costa (1985)
($Q_{max} = 1.122 V_s^{0.57}$, where $V_s$ is the barrier lake volume), which produced a result of 2273
m$^3$/s that is close to the result obtained by Manning's equation.
The temporal distributions of flood depth, velocity, and shear stress in the two scenarios are
presented in Fig. 6. The flood completely submerged all buildings on the left bank near the
middle of the river channel, and the buildings on the river terrace on the right bank were strongly
eroded. The maximum water depth and velocity of the dam-burst flood were 13.96 m and 8.24
m/s, respectively, which were 1.24 and 1.31 times higher than those of the ordinary flood,
respectively. The maximum shear stress of the flood in the main channel increased sharply from
320 Pa to 853 Pa, indicating a 2.67-fold increase compared to the ordinary flood. For the ordinary
flood scenario, the water depth and velocity were high in the channel and decreased in the
floodplain. In contrast, the high velocity and shear stress zones that resulted from the dam-burst
flood were mainly distributed in the main channel and along the left bank, indicating that the
material deposited by the debris flow and the original river bank are highly susceptible to erosion.

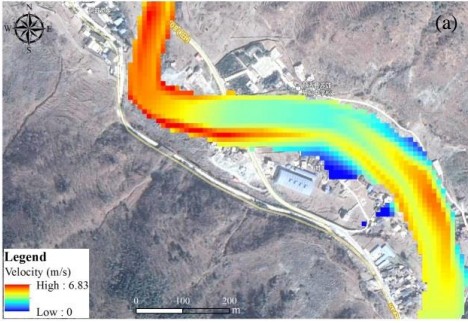
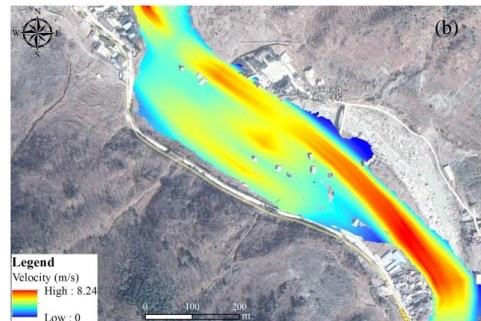
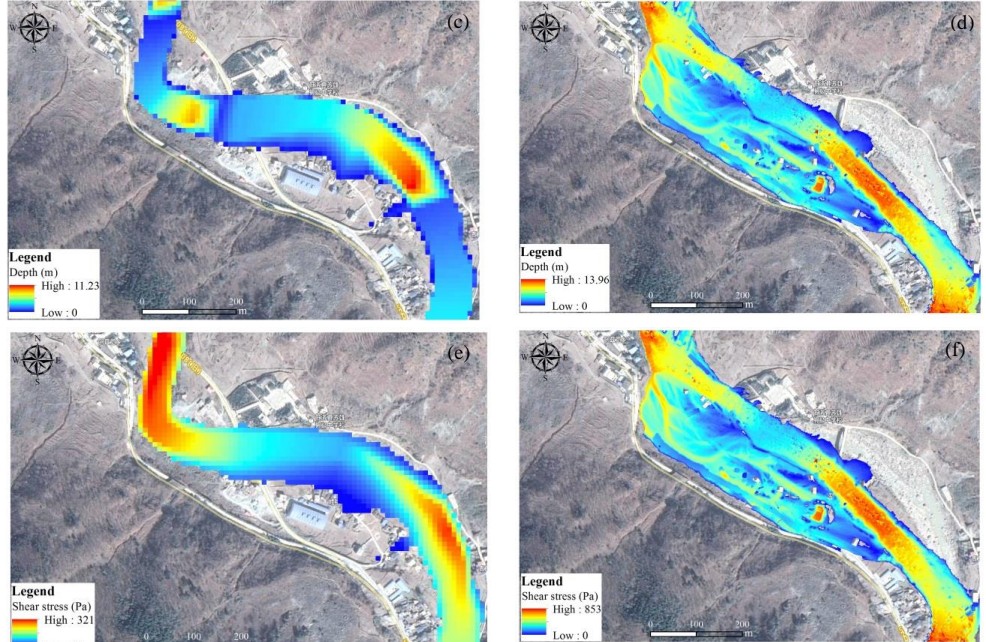

**Figure 6** Distribution of depth, velocity, and shear stress of ordinary flood and dam-burst flood: (a) Maximum velocity distribution of ordinary flood;(b)Maximum velocity distribution of dam-burst flood;(c) Maximum depth distribution of ordinary flood;(d) Maximum depth distribution of dam-burst flood;(e) Maximum shear stress distribution of ordinary flood;(f) Maximum shear stress distribution of dam-burst flood.

The critical shear stress for bedload transport in the gravel-bed river is determined by the equation $\theta = \frac{\tau}{(\rho_s - \rho)gD}$ =0.04, where $\theta$ is the critical shear stress, $\tau$ is the bed shear stress, $\rho_s$ is the soil mass density, $\rho$ is the water mass density, $g$ is the gravitational acceleration, and D is the sediment diameter (Petit et al., 2015). The dam-burst flood had the potential to transport large boulders up to 1.3 m in diameter, while an ordinary flood could only move gravel up to 0.49 m in diameter. Such high shear stress also demonstrated the strong erosional ability of the dam-burst flood, which seriously scoured the debris sediment deposit and original riverbank, transporting coarse gravel and forming a new straight river channel. The new channel is straighter and steeper than the original channel, raising the bed of the Niri River by 1-17 m and burying buildings up to 1 km downstream of Heixluo Gully. The channel length shortened from 1010 m to 842 m, and the channel gradient increased from 1.71% to 2.72%. The change in the river channel led to an inundation area that deflected to the left. Buildings built on the original left riverbank were first impacted by debris flow and subsequently destroyed by the flood. The river terrace on the original right bank was strongly eroded by the flood, leading to the collapse and demolition of buildings. Five river sections (Section 1 to Section 5) were selected to analyze the terrain changes

(see Fig. 4). From Section 1 to Section 3, the main channel varied from the right bank to the left
bank with a distance between 40 m and 100 m, the average width of the new river channel was 50
m, and the vertical distance between the new riverbed and floodplain was 11.23 m. In Section 5,
the channel migrated from the left bank to the right bank due to the severe erosion of the original
river terrace and had a maximum depth of 10 m (Fig. 7). The channel width increased to
approximately 100 m, and the channel depth decreased to less than 5 m.

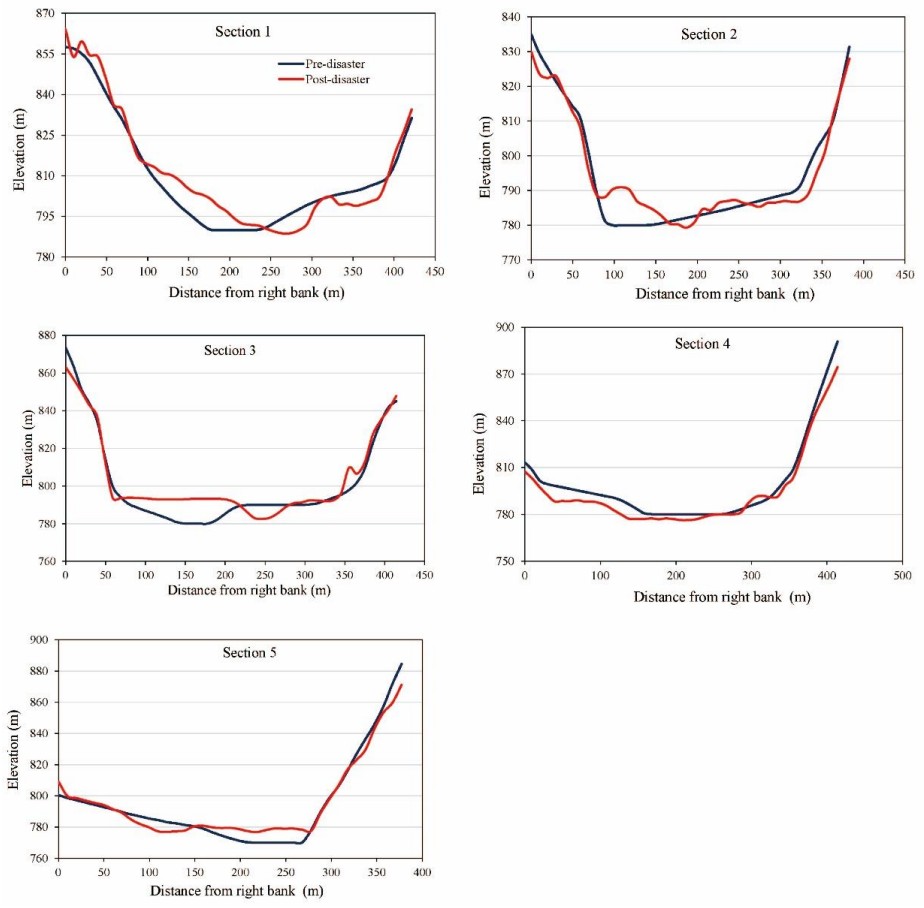


**Figure 7** Cross-section profile before and after the disaster.

## 5.3 Damage patterns of buildings

The debris flow-outburst flood hazard chain caused damage to 89 buildings, a 1.2 km stretch of
national road G245, and 5 highway bridges along the main river. The evolution of this hazard
chain occurred in two phases. First, the debris flow blocked the main river and formed a barrier
dam and dammed lake, which was, second, followed by the outburst of the lake that led to the





subsequent flooding and inundation. During the first phase, a significant amount of sediment was
transported by the debris flow to the confluence area and deposited in the river channel, which
formed a barrier lake with a volume of 857,504 m$^3$. The barrier lake breached completely only
approximately 30 minutes later, leading to a highly energetic flood that caused serious erosion of
the riverbank and the formation of the outburst flood, a new straight river channel.
Fig. 8 illustrates the boundary of debris flow deposition and dam-burst flood inundation. The
debris flow deposition boundary was determined via a field investigation and compared with
results obtained from a liquid–solid two-phase model, which takes into account the effects of
runoff and entrainment (Li et al., 2022). The flooding boundary was obtained by combining the
results of the HEC-RAS simulation with field survey data. The confluence area was heavily
impacted by the debris flow, resulting in the transportation of a significant amount of solid
materials over an area of 0.157 km$^2$. As a result, the majority of the village's buildings were
inundated by the debris flow. The dam-burst flood caused serious damage to buildings by
flushing a large volume of debris flow sediment and riverbank material downstream.
Three hazard zones are identified based on the boundary of the debris flow and dam-burst
flood, as shown in Fig. 8 and Fig. 9. The damage patterns of buildings in the different hazard
zones can be classified into three categories, namely, (I) buildings only buried by debris flow; (II)
buildings only inundated by dam-burst flood; and (III) buildings sequentially buried by debris
flow and inundated by dam-burst flood. Zone (I) is situated near the Heixiluo gully mouth, where
the debris flow transported a large volume of sediment and seriously eroded the sidewall and bed
of the channel, expanding the channel's width from 10 m to 40 m. All buildings were inundated
by sediment to a depth of over 6 m.
Zone (II) is subdivided into two subzones, Zone (II) ① and Zone (II) ②, based on the spatial
location. Zone (II) ① is situated in the upstream reach of the Niri River, near the debris flow dam,
and is mainly inundated by the static water of the dammed lake (Fig. 9(b)). Zone (II) ② lies on the
right bank of the downstream reach of the Niri River, outside the debris flow fan. The original
right riverbank in Zone (II) ② was a terrace 10 m high that was severely scoured by the highly
energetic flood with a shear stress greater than 450 Pa. The entire terrace was cut off, and a new
channel was formed across the middle area (Fig. 9(c)). The erosion area on the river terrace
measures approximately 1800 m$^2$ with a length of 300 m and a width of 60 m. Two buildings
situated on the upper part of the river terrace collapsed and disintegrated due to the impact of the
flood (part (a) in Fig. 9(d)). A three-story building was partially destroyed due to foundation
erosion. The buildings on the lower part of Zone (II) ② were simultaneously buried by the
sediment transported by floods and inundated by floodwater (part (b) in Fig. 9(d)).
Zone (III) is primarily located on the left bank of the original river and the lower part of the
debris flow fan. The original river channel is filled with debris up to a depth of 10 m. The debris
flow transported sediment across the raised riverbed into villages and formed a slope that was
high on the right and low on the left in the confluence area. Then, the flood breached the debris
flow dam and severely eroded the deposited debris and the original floodplain surface, resulting
in a new straight channel. The buildings on the left bank of the river, which were buried by the
debris flow, were sequentially impacted by the dam-breach flood. The flood heavily damaged
buildings near the new river channel, and floodwater from the channel was observed to always
inundate the buildings. Notably, the boundaries of the different damage zones are not static. The
extent of the damage zone is not the same for other confluence areas; it is determined by the
dynamic characteristics of hazards and is also influenced by the local terrain.

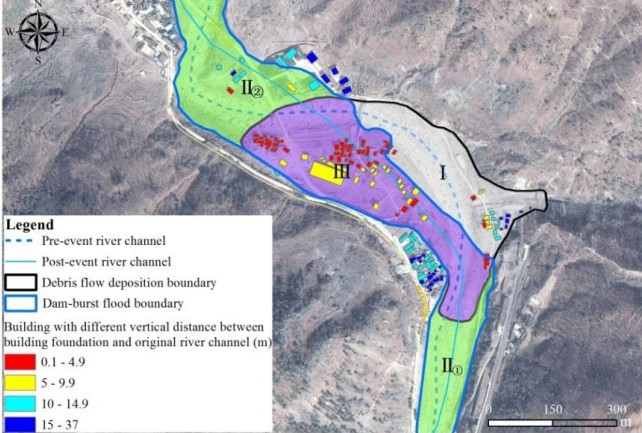

**Figure 8** Inundation boundary of debris flow and dam-burst flood and spatial division of the hazard zone
based on building damage patterns: (I) buried by debris flow; (II) inundated by dam-burst flood; (III)
buried by debris flow and the inundated by dam-burst flood.
A total of 110 buildings in the village were impacted by the multi-hazards, accounting for
69.62% of the total buildings. Among them, 75 buildings located in Zone (III) were impacted by
the debris flow and flood in succession, which accounted for 47.47% of the total buildings. In
contrast, buildings destroyed by the debris flow in Zone (I) and dam-burst flood in Zone (II)
accounted for only 16.46% and 5.70% of the total buildings, respectively (Table 3). Overall, the
number of buildings within the debris flow deposition boundary and flood inundation boundary is
101 and 84, respectively, accounting for 63.92% and 53.16% of the total buildings in the village.


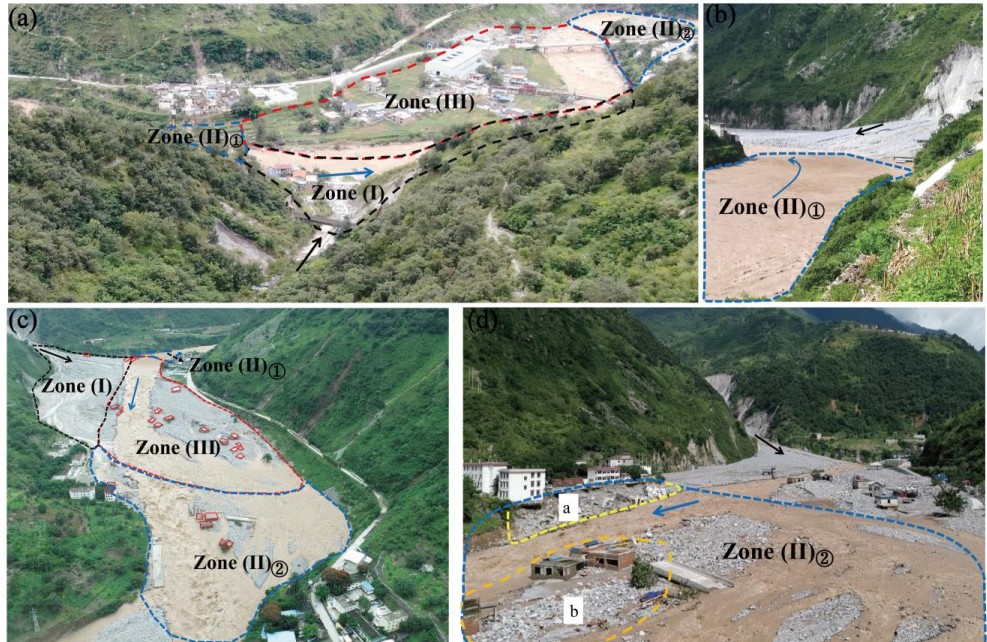

**Figure 9** Spatial distribution of the three hazard zones before and after the disaster: (a) before the disaster; (b) (c) (d) after the disaster.

**Table 3** Statistics of buildings damaged by the debris flow and dam-burst flood

| Damage pattern | (I) Buried by debris flow | (II) Inundated by dam-burst flood | (III) Buried by debris flow and inundated by dam-burst flood sequentially | Sum |
|---|---|---|---|---|
| Total number of buildings destroyed | 26 | 9 | 75 | 110 |
| The proportion of damaged buildings to the total buildings in the village (%) | 16.46 | 5.70 | 47.47 | 69.62 |

The impact force of fluvial sediment transport is greatly influenced by the relative distance of buildings to channels (Wei et al., 2022). Buildings that are close to the channel are always more vulnerable to damage than those located farther away from the river. During the hazard chain, a total of 84 buildings in Zone (II) and Zone (III) were impacted by the dam-burst flood (Fig. 10). To assess the influence of building distance from the river channel, we analyzed the vertical distances between the damaged building foundation and the original river channel based on pre-event terrain (Table 4). We found that 67.86% of all damaged buildings were within 5 m of the channel, while 23.81% of all damaged buildings were between 5 m and 10 m of the original channel. Buildings that were located at distances greater than 10 m only accounted for 8.33% of



the total damaged buildings. In contrast, the average vertical distance of undamaged buildings
was 15.3 m, with a minimum value of 11.4 m.
**Table 4** Statistics of the vertical distance between the damaged building foundation and
original river channel within the whole flooding boundary

| The vertical distance between the building foundation and original river channel (m) | (0, 5) | (5,10) | (10,16) | Sum |
|---|---|---|---|---|
| Total number of buildings destroyed | 57 | 20 | 7 | 84 |
| The proportion of damaged buildings to the total (%) | 67.86 | 23.81 | 8.33 | 100 |

**5.4 Vulnerability analysis of the buildings**
Most of the buildings in the village were completely buried by sediments or collapsed with no
visible remains. To construct vulnerability curves, 29 damaged buildings with brick-concrete
structures located in the three hazard zones were selected (Fig. 9(c), Fig. 10). Of these, 2
buildings were located in Zone (I), 7 buildings were located in Zone (II), and the rest were
distributed throughout Zone (III).
The building characteristics and hazard intensity are presented in Table 5. The two buildings
in Zone (I) were buried by debris flow with a thickness greater than 5 m. In Zone (III), buildings
located near the debris flow dam (such as buildings 3, 4, and 5) were first buried by the debris
flow and then inundated by water from the dammed lake for 30 minutes. These buildings were
then impacted by the dam-burst flood. Additionally, buildings near the new river channel suffered
greater impact pressure than other buildings. For example, the residual broken structures of
buildings 7 and 8 were heavily damaged by the direct impact of the flood in the vertical direction.
The walls of the two buildings were severely abraded by impact pressures of 75.7 kPa and 71.1
kPa, respectively. Additionally, the foundations of the two buildings were partially scoured by
floods with high shear stresses of 562 Pa and 553 Pa, respectively.
Buildings located in Zone (II) were only severely impacted by the dam-burst flood. For instance,
the foundation of the three-story school building (building 27) was severely eroded by the flood
to a scour depth of 1 m, and the floors on the right were collapsed. There was no evidence on the
walls of the building that the debris flow had abraded the structure. The velocity and shear stress
of the flood in this location were 5.3 m/s and 463 Pa, respectively. Buildings 23-27, which were
close to the new river channel, were thoroughly buried by the sediment transported by the flood
and inundated by floodwater.
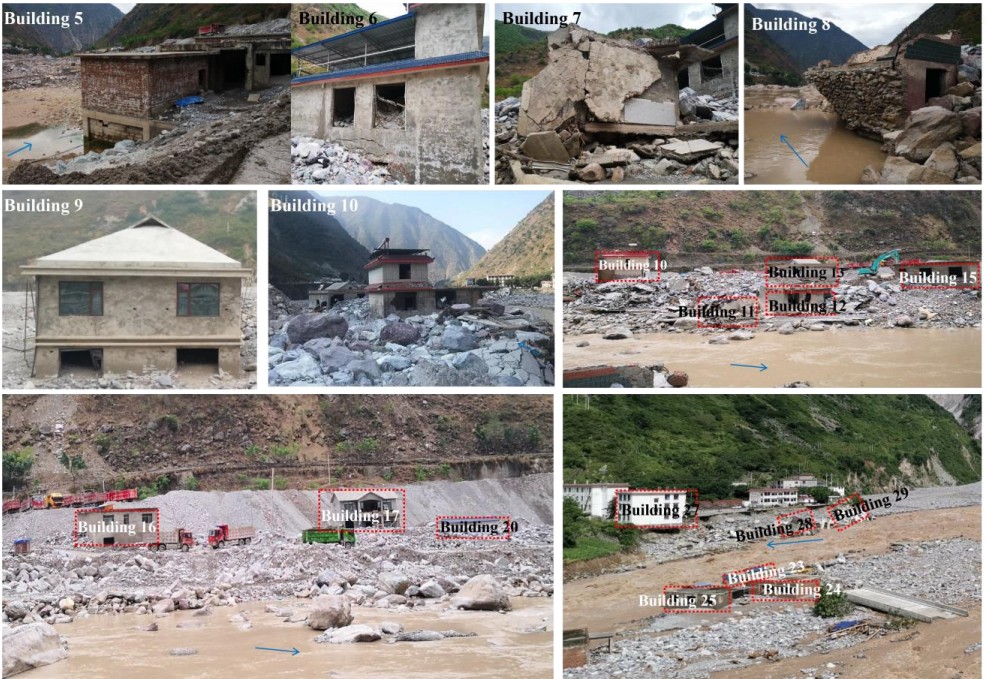

**Figure 10** Buildings with the different degree of damages within three hazard zones.

The vulnerability curve in Zone (II) and Zone (III) was developed by summing up the damage caused by the multiple hazards and impact pressure (Fig. 11). Logistic functions were proposed separately for the two hazard zones, and the corresponding determination coefficient ($R^2$) and root mean square error (RMSE) were also obtained. The determination coefficients of the two regression curves in Zone (III) have a higher $R^2$. The RMSEs of the curves in Zone (II) and Zone (III) are 0.08 and 0.14, respectively. The correlation between vulnerability and inundation depth in the two zones is shown in Fig. 12, with an $R^2$ lower than impact pressure ($R^2$=0.30 for Zone (II) and $R^2$=0.41 for Zone (III)). Building vulnerability increases with increasing hazard intensity, and the trend is similar in the two zones. For the same impact pressure and inundation depth, the damage to buildings in Zone (II) is greater than that in Zone (III). The threshold of the impact pressure in Zone (II) and Zone (III) where vulnerability is equal to 1 is 88 kPa and 106 kPa, respectively, which is much greater than that of the three functions.

**Table 5** Database of the damaged buildings

| Building | Debris flow deposition depth (m) | Debris flow velocity (m/s) | Debris flow impact pressure (kPa) | Flood depth (m) | Flood velocity (m/s) | Flood impact pressure (kPa) | Damage ratio | Hazard zone |
|---|---|---|---|---|---|---|---|---|
| 1 | 5 | 3.9 | 39.7 | | | | 0.9 | I |




| | | | | | | | | |
|---|---|---|---|---|---|---|---|---|
| 2 | 6 | 4.1 | 46.0 | | | | 1 | I |
| 3 | 1 | 2.6 | 11.8 | 1.2 | 1.0 | 7.0 | 0.7 | III |
| 4 | 2 | 3.1 | 19.4 | 1.3 | 2.0 | 10.3 | 0.6 | III |
| 5 | 2 | 3.1 | 19.4 | 1.3 | 2.3 | 11.6 | 0.6 | III |
| 6 | 7 | 4.2 | 52.2 | 3.7 | 4.3 | 36.8 | 0.8 | III |
| 7 | 1.5 | 4.1 | 23.9 | 6.7 | 6.5 | 75.5 | 1 | III |
| 8 | 6 | 3.4 | 41.2 | 6.3 | 6.3 | 71.1 | 1 | III |
| 9 | 5 | 3.9 | 39.7 | 1.0 | 2.0 | 8.8 | 0.7 | III |
| 10 | 3 | 3.4 | 26.5 | 2.1 | 4.1 | 27.3 | 0.6 | III |
| 11 | 2 | 3.1 | 19.4 | 6.4 | 6.7 | 76.6 | 1 | III |
| 12 | 2 | 3.1 | 19.4 | 6.3 | 6.0 | 67.1 | 1 | III |
| 13 | 2 | 3.1 | 19.4 | 0.9 | 3.6 | 17.6 | 0.7 | III |
| 14 | 4 | 3.7 | 33.2 | 4.4 | 5.9 | 56.3 | 1 | III |
| 15 | 5 | 3.9 | 39.7 | 3.6 | 5.1 | 43.3 | 0.7 | III |
| 16 | 0.5 | 2.2 | 7.3 | 0.7 | 1.5 | 5.8 | 0.4 | III |
| 17 | 1 | 2.6 | 11.8 | 1.2 | 0.8 | 6.5 | 0.3 | III |
| 18 | 0.5 | 2.2 | 7.3 | 3.0 | 4.6 | 35.7 | 1 | III |
| 19 | 0.5 | 2.2 | 7.3 | 3.9 | 5.0 | 44.4 | 1 | III |
| 20 | 1 | 2.6 | 11.8 | 2.4 | 3.8 | 26.0 | 0.7 | III |
| 21 | 1 | 2.6 | 11.8 | 2.4 | 4.1 | 28.1 | 0.9 | III |
| 22 | 0.5 | 2.2 | 7.3 | 3.5 | 4.7 | 39.5 | 1 | III |
| 23 | | | | 5.3 | 5.4 | 55.3 | 0.8 | II |
| 24 | | | | 1.6 | 3.2 | 18.2 | 0.7 | II |
| 25 | | | | 4.7 | 4.9 | 47.2 | 0.8 | II |
| 26 | | | | 3.7 | 3.6 | 31.5 | 0.7 | II |
| 27 | | | | 3.7 | 5.3 | 45.8 | 0.9 | II |
| 28 | | | | 4.5 | 4.4 | 41.0 | 1 | II |
| 29 | | | | 5.1 | 5.1 | 51.4 | 1 | II |


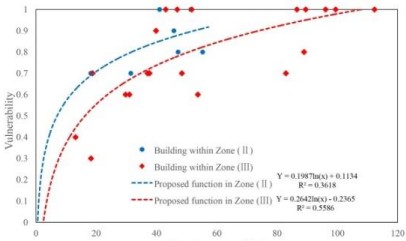

**Figure11** Proposed vulnerability functions based on the impact pressure in Zone (II) and Zone (III).

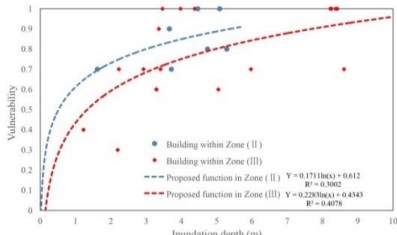

**Figure12** Proposed vulnerability functions based on the inundation depth in Zone (II) and Zone (III).




The vulnerability curves proposed for Zone (II) and Zone (III) were compared to the three
functions used in debris flow risk assessment (Fig.13, Fig.14). The functions developed by Quan
et al. (2011) and Kang et al. (2016) were calculated based on damage done to brick masonry and
nonreinforced concrete structures that had been impacted by the debris flows in South Korea and
Italy, respectively. The vulnerability curve proposed by Zhang et al. (2018) was developed for
buildings with brick-concrete structures from the Zhouqu debris flow event in China. The slope
of the two proposed vulnerability curves based on impact pressure is smaller than those of the
three curves. When the impact pressure is less than 20 kPa, the proposed curves show a similar
increasing trend compared to the three functions. However, when the impact pressure is greater
than 20 kPa, the slope of the two proposed vulnerability curves is much smaller than those of the
three curves. For the curves based on inundation depth, when the depth is less than 2 m, the slope
is steeper than that of Quan et al. (2011) and Zhang et al. (2018) and slower than that of Kang et
al. (2016). When the depth is greater than 2 m, the damage increases slower than the curves of
Quan et al. (2011) and Zhang et al. (2018). This disparity may be attributed to the different
damage patterns and structures of the buildings in this study. The three vulnerability functions
were generated for a single debris flow event, whereas the mechanisms by which buildings
impacted by floods fail is not the same when those buildings are subjected to a debris flow. The
structures of most buildings in the study area are tougher than those in the three events, and
nearly half of the buildings had been recently built by a more professional construction team. For
example, the newly built four buildings 6, 9, 10, and 15 were not completely damaged by hazard
chain under impact pressures greater than 40 kPa.

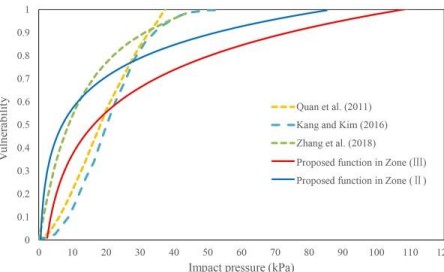

**Figure 13** Comparison of the building vulnerability functions with the impact pressure functions proposed by Quan et al. (2011), Kang et al. (2016), and Zhang et al. (2018).

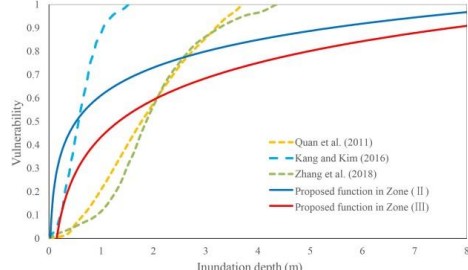

**Figure14** Comparison of the building vulnerability functions with the inundation depth proposed by Quan et al. (2011), Kang et al. (2016), and Zhang et al. (2018).

The building damage distribution chart shows building damage plotted as a function of debris
flow and flood impact pressure (see Fig. 15). The figure includes aggregated damage to buildings


impacted by the sequentially occurring hazards in Zone (III) and damage caused by a single
hazard in Zone (I) and Zone (II). Damage is divided into three categories based on the threshold
impact pressure: slight damage (0.3-0.4), moderate damage (0.6-0.7), and heavy and complete
damage (0.8-1.0). Heavy damage occurs at impact pressures greater than 40 kPa, while slight
damage occurs below 20 kPa. Moderate damage occurs at impact pressures between 20 kPa and
40 kPa. The threshold impact pressure is compared with that proposed by Hu et al. (2012) and
Zanchetta et al. (2004), which were derived from a single debris flow disaster in China and Italy,
respectively. Although the detailed definition of the damage scales differs, the threshold of the
impact pressure for buildings at the slight, heavy, and complete damage scales is generally larger
than that for the brick-concrete structures presented in Hu et al. (2012) and smaller than that for
the reinforced concrete frames also presented in Hu et al. (2012) and the masonry structures with
basements presented in Zanchetta et al. (2004). A similar trend for the threshold of the impact
pressure for buildings with a moderate damage scale can be observed.

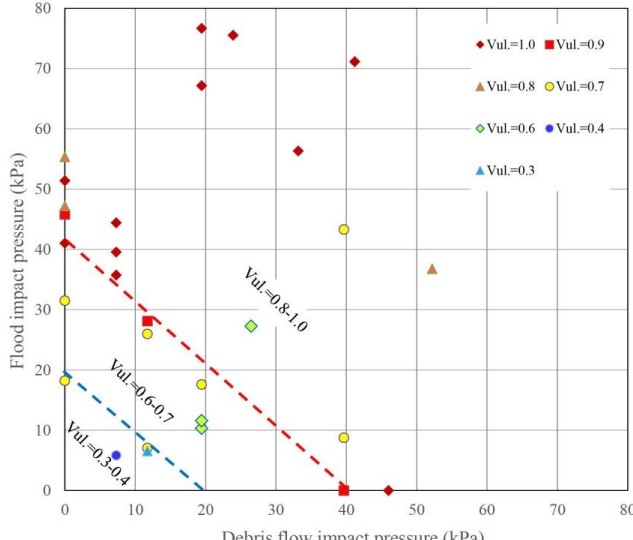


**Figure 15** Accumulation of building damage due to debris flow and dam-burst flood. The damage
distribution is based on the debris flow and flood impact pressure (Vul. refers to vulnerability).

The building damage distribution chart remains a valid tool for assessing the vulnerability of
buildings affected by debris flows and flash floods, despite not incorporating all damage ratios.
However, some limitations and uncertainties exist within the vulnerability functions. For instance,
calculating a single average impact pressure value prebuilding for building clusters introduces
uncertainty, as water depth and velocity differ significantly at different sides of the building due


to the shielding effect (Hu et al., 2012; Arrighi et al, 2020). Furthermore, the building's geometry,
direction, orientation, and maintenance condition are not considered in the vulnerability analysis.
The amplification of debris flow damage is due to subsequent flooding in time and space.
Aggregated damage (i.e., damage caused by both debris flows and floods) is applied in the
vulnerability analysis for areas that are successively struck by debris flows and floods. However,
the amplified damage effect of the dam-burst flood on debris flow was not accurately quantified
because of the absence of a database containing information regarding the damage done by the
debris flow before the dam burst. As a result, more detailed data are needed to assess the
cumulative impact of hazard chains on building vulnerability.

## 6 Discussion

### 6.1 Expansion of flood inundation extent

As a result of the confluence zone's location on a river bend, the dam-burst flood typically
flows in a straight direction and creates a new straight channel when the river channel becomes
completely blocked. This channel translocation leads to a larger flooded area and causes more
severe damage to buildings on the floodplain. The flood inundation zones in the village expanded
to $110^5$ m$^2$, which is up to 4 times the area of an ordinary flood due to flood amplification (Fig.
16). In the expanded inundation zone, 41 buildings, a traffic road spanning 410 m, and farmland
with an area of $10 \times 10^4$ m$^2$ were submerged. The buildings located in the middle of the inundation
zones suffered the most severe damage due to the floodwater's high scouring capability and
sediment transport capacity. Many buildings near the flow collapsed, and most structures were
carried away by the water current.
Table 6 presents a comparison of the dynamic characteristics and damage increments between
ordinary and dam-burst floods in different locations. The damage increment is calculated based
on the proposed function in Zone II and is the ratio of the damage caused by the two floods.
Buildings 6, 7, 8, 14, and 15 were situated close to the new river channel, and the bed shear stress
and impact pressure increased up to 24.67 times and 6.76 times that of an ordinary flood,
respectively. Building 21 was constructed near the original channel, and the increase of bed shear
stress and impact pressure was smaller than in the other buildings. Overall, the average bed shear
stress and impact pressure increased by 16.83 and 4.24 times, respectively, due to flood
amplification. The average damage to the six buildings located near the new channel increased by
119% due to the lake created by the debris flow barrier.


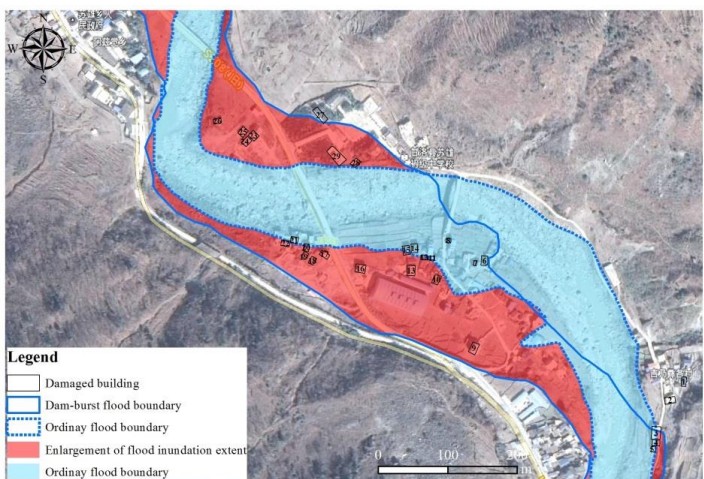

**Figure16** The inundation extent of ordinary floods and dam-burst floods.

**Table 6** Comparison of dynamic characteristics and degree of damage between ordinary floods and dam-burst floods in different locations

| Location | The ratio of dam-burst flood to ordinary flood | | | | |
|---|---|---|---|---|---|
| | Depth | Velocity | Bed shear stress | Impact pressure | Damage degree |
| Building 6 | 2.00 | 1.78 | 9.88 | 2.46 | 1.19 |
| Building 7 | 3.49 | 3.20 | 24.67 | 5.54 | 1.36 |
| Building 8 | 3.12 | 2.14 | 17.45 | 3.79 | 1.26 |
| Building 14 | 6.49 | 2.64 | 24.43 | 6.76 | 1.45 |
| Building 15 | 6.30 | 2.20 | 18.58 | 5.34 | 1.40 |
| Building 21 | 2.95 | 1.08 | 5.96 | 1.55 | 1.09 |
| Average value | 4.05 | 2.17 | 16.83 | 4.24 | 1.19 |

**6.2 Damage aggravation due to hazard chain**

The debris flow-outburst flood hazard chain extends the damage in time and space. The blockage of the river results in upstream inundation and downstream outburst flood, exacerbating the damage caused by the debris flow. To assess the amplified damage effect on building vulnerability, impact pressure functions proposed by Quan et al. (2011) and Zhang et al. (2018) were applied to calculate the damage caused by the debris flow. Table 7 presents a comparison of the calculated damage with the aggregated damage caused by the debris flow and outbreak flood. Except for buildings 6, 9, 10, and 15, the damage to buildings increased markedly due to the cumulative effect that the dam-burst flood had on building vulnerability. On average, the damage increased by 4.8 times and 5.4 times compared to the damage results calculated by Quan et al.




(2012) and Zhang et al. (2018), respectively. The aggravated damage is up to 20 times that caused
by debris flow alone.
**Table 7** Comparison of damage caused by debris flow with the cumulative damage

| Building | Debris flow inundation depth (m) | Aggregated damage ratio (D1) | Damage ratio calculated by Quan et al. (2012) (D2) | Damage ratio calculated by Zhang et al. (2018) (D3) | Ratio of D1 to D2 | Ratio of D1 to D3 |
|---|---|---|---|---|---|---|
| 3 | 1 | 0.7 | 0.2 | 0.12 | 3.5 | 5.8 |
| 4 | 2 | 0.6 | 0.57 | 0.56 | 1.1 | 1.1 |
| 5 | 2 | 0.6 | 0.57 | 0.56 | 1.1 | 1.1 |
| 6 | 7 | 0.8 | 1 | 1 | 0.8 | 0.8 |
| 7 | 1.5 | 1 | 0.4 | 0.3 | 2.5 | 3.3 |
| 8 | 6 | 1 | 1 | 1 | 1 | 1 |
| 9 | 5 | 0.7 | 1 | 1 | 0.7 | 0.7 |
| 10 | 3 | 0.6 | 1 | 0.86 | 0.6 | 0.7 |
| 11 | 2 | 1 | 0.57 | 0.56 | 1.8 | 1.8 |
| 12 | 2 | 1 | 0.57 | 0.56 | 1.8 | 1.8 |
| 13 | 2 | 0.7 | 0.57 | 0.56 | 1.2 | 1.3 |
| 14 | 4 | 1 | 1 | 0.96 | 1 | 1.0 |
| 15 | 5 | 0.7 | 1 | 1 | 0.7 | 0.7 |
| 16 | 0.5 | 0.4 | 0.05 | 0.05 | 8 | 8 |
| 17 | 1 | 0.3 | 0.2 | 0.12 | 1.5 | 2.5 |
| 18 | 0.5 | 1 | 0.05 | 0.05 | 20 | 20 |
| 19 | 0.5 | 1 | 0.05 | 0.05 | 20 | 20 |
| 20 | 1 | 0.7 | 0.2 | 0.12 | 3.5 | 5.8 |
| 21 | 1 | 0.9 | 0.2 | 0.12 | 4.5 | 7.5 |
| 22 | 0.5 | 1 | 0.05 | 0.05 | 20 | 20 |

**6.3 The implication of hazard mitigation**
In recent years, the hazard chain of debris flows and outburst floods has become more frequent
in high mountain regions due to the impact of climate change and earthquakes (Chen et al., 2022).
The damage caused by the primary debris flow can be intensified and enlarged due to the
successive dam-burst flood.
Risk assessment for debris flow-outburst flood hazard chains is crucial to mitigate the damage
posed to structures in the confluence zone. Risk analysis should incorporate both the debris flow
initiation mechanism and the mechanism that generates the dam-burst flood (Chen et al., 2022). A
detailed investigation should be conducted for the exposed elements in the confluence zone and in


both the upstream and downstream reaches of the river. Based on the disaster transformation process and the failure mechanisms of structures, hazard zones should be identified, and corresponding disaster reduction measures should be developed (Cui and Guo, 2021). Moreover, specific structural measures are urgently needed. First, engineering measures should be implemented in the watershed to mitigate debris flows (Cui and Lin, 2013). Second, buildings should not be constructed near debris flow gullies, and new buildings should be built on elevated ground or at certain elevations above the ground (Attems et al., 2019). Third, deflection walls should be considered and constructed in villages susceptible to debris flows to protect entire buildings (Wang et al., 2022), and flood protection walls should be built along the main river to protect the entire flood-prone village.

## 7 Conclusions

Buildings in the confluence zone of a debris flow-prone catchment and along a main river channel are highly vulnerable to a debris flow-dam-burst flood hazard chain. Assessing building damage is essential for risk mitigation and resilient construction. However, research concerning building damage mainly focuses on a single debris flow or flash flood and fails to consider the different damage characteristics of buildings exposed to both hazards simultaneously. Therefore, studying the characteristics and patterns of building damage in confluence areas can help to develop a reliable vulnerability assessment method. In this study, we investigate the dynamic characteristics of the hazards and damage patterns of the 2020 Heixiluo debris flow and dam-burst flood disaster. We draw the following conclusion:

1. The dam-burst flood, which had a peak discharge of 2,273 $m^3/s$, seriously eroded the debris flow fan and formed a new straighter and steeper channel. The maximum estimated velocity was 8.24 m/s, and the bed shear stress reached 853 Pa. The flood's inundation extent in the confluence zone was expanded by a factor of 4, and the impact pressure increased up to 6.76 times due to flood amplification.

2. The damage patterns of the buildings were classified into three types: (I) buried by primary debris flow, (II) inundated by secondary dam-burst flood, and (III) buried by debris flow and inundated by dam-burst flood sequentially. Three corresponding hazard zones were identified. The damage to buildings in Zone (III) was 20 times more intense due to the dam-burst flood. The spatial division of hazard zones can be applied to the selection of building sites and to the planning of structural measures in the confluence area.

3. The vulnerability curves show a similar increasing trend with impact pressure and inundation depth in Zones II and III, and the threshold of the impact pressures in Zones II and III





where vulnerability is equal to 1 are 88 kPa and 106 kPa, respectively. A vulnerability assessment
chart was developed, and three categories, namely, slight damage (0.3-0.4), moderate damage
(0.6-0.7), and heavy and complete damage (0.8-1.0), were identified. Heavy damage occurs at an
impact pressure greater than 40 kPa, while slight damage occurs below 20 kPa. Moderate damage
occurs at an impact pressure between 20 kPa and 40 kPa.
4. Some uncertainties and limitations are involved in vulnerability analysis. The building's
physical characteristics, such as shape, orientation, and maintenance condition, should be
considered for the vulnerability analysis. Further investigation and research are  recommended to
explore the cumulative effect of multiple hazards on building vulnerability. Despite the
deficiencies, vulnerability curves and assessment charts are valuable to analyze the risk posed by
debris flow hazard chains within the confluence zone.

## Acknowledgements

This work has been financially supported by the the Second Tibetan Plateau Scientific Expedition
and Research Program (2019QZKK0902) and the National Natural Science Foundation of China
(41790434).


## Data availability

All raw data can be provided by the corresponding authors upon request.

## Author contributions

Kaiheng Hu contributed to the conception of the study; Li Wei performed the data analyses and
wrote the manuscript draft; Shuang Liu performed the data analyses. Nan Ning, Xiaopeng Zhang
and Qiyuan Zhang performed the field investigation; Md Abdur Rahim reviewed and edited the
manuscript.

## Competing interests.

The authors declare that they have no conflictof interest.




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
