# Peer review of "The vulnerability of buildings to a large-scale debris flow and outburst flood hazard cascade that occurred on 30 August 2020 in Ganluo, Southwest China"

_Natural Hazards and Earth System Sciences, 2023_

## Author Response (AR1)

Dear Editors and Reviewers:

On behalf of my co-authors, we thank you very much for giving us an opportunity to revise our manuscript, we appreciate editor and reviewers very much for their positive and constructive comments and suggestions on our manuscript. We have studied the comments from reviewers, and we have tried our best to revise our manuscript according to the comments.

We have made the following major changes in this article:

1. Debris flow simulation was added, and the description of debris flow simulation methods and simulation results were added in the article. 2. Then, the article structure has been adjusted as follow: 1 Introduction, 2 Study area, 3 Data and methods (3.1 Data collection, 3.2 Methodology), 4 Results (4.1 Hazard cascade, 4.2 Dynamic characteristics of the debris flow, 4.3 Dynamic characteristics of the outbreak flood, 4.4 Damage patterns of buildings, 4.5 Vulnerability analysis of the buildings) 5 Discussion, 6 Conclusion..

3. The intensity characteristics and inundation range of debris flow in the original draft were determined according to field investigation.The conclusion drawn from field investigation were deleted in the origin draft.

5 The chart was redrawn, as shown in Figure 1, Figure 2, Figure 7, Figure 9, etc.

In addition, The main corrections in the paper and the responds to the reviewer's comments are as flowing.

| Reviewer 1 | Comments | Response |
|---|---|---|
| 1 | It is a multi-hazard situation and you (if I understood well) made only simulations of the final flood event, neglecting the debris flow part. So, you cannot really say that you analysed the full chain as you consider the first part as known and set as a basis. | We have supplemented the simulation results of debris flow, and the subsequent analysis is also based on the dynamic characteristics of the simulated debris flow (**see Page 6 and L. 158-166,see Page 11 and L. 234-246,Figure 6**) |

| | | |
|---|---|---|
| 2 | The presentation of the maps in general, also those including flood simulation results are not very clear - so, some more work is necessary to communicate your results to others. | We redrew the picture using clearer remote sensing image. **(see Figure 6, Figure 7, Figure 9, Figure 17)** |
| 3 | Therefore, right from the beginning you should indicate that the core analysis is focused on the final part of the chain, taking the first part as a given element. This will require a restructuring of the paper and thus a major revision | First, we added the debris flow simulation part, and all the analyses were based on the debris flow simulation results (**see P 6 and L. 158-166,see P 11 and L. 234-246**). Then, the article structure has been adjusted as follow: 1 Introduction, 2 Study area, 3 Data and methods (3.1 Data collection, 3.2 Methodology), 4 Results (4.1 Hazard cascade, 4.2 Dynamic characteristics of the debris flow, 4.3 Dynamic characteristics of the outbreak flood, 4.4 Damage patterns of buildings, 4.5 Vulnerability analysis of the buildings) 5 Discussion, 6 Conclusion. |

| Reviewer 2 | Comments | Response |
|---|---|---|
| 1 | Does the parameters of model or data representative in this case study considering different scales? such as rainfall data, some parameters of models in Table1, or three functions used in debris flow risk assessment. how well does the models' results? | The formula in Table 1 is an empirical formula, and the parameters in it are based on values that are taken from surveys or experience, so it will not take into account the influence of rainfall and other factors. |
| 2 | A spatial distribution map of damage might be more clearly show the distance from river channel, also the spatial relationship between vulnerability and factors might be more useful. Besides, does some field investigation or actual evidence that can support the vulnerability assessment? | The spatial distribution map of buildings is shown in Figure 9 and Figure 17. In addition, because our vulnerability analysis is based on dynamic process simulation and field investigation, the results of the analysis are mainly suitable for the study area of this paper, and have not been verified in other areas. |
| 3 | This vulnerability assessment mainly focus on the buildings, is it possible to combine some other factors to analysis, so that might be more comprehensive. | Roads and cultivated land in the region were also severely damaged, but it is difficult to obtain the details of their damages in the post-disaster survey due to |

| | | | |
|---|---|---|---|
| | | post-disaster rescue excavation, etc., so only the buildings are analyzed in detail in this paper. | |

| Reviewer 3 | Comments | Response |
|---|---|---|
| 1 | I think it is more appropriate to use "hazard cascade" instead of "hazard chain". Consider modifying this in the title and the whole manuscript. | Thanks for your advice, we use "hazard cascade" in the whole manuscript. The reference (Cutter, 2018) was cited in the manuscript. |
| 2 | L15. For this type of process, it is also more common to refer to it as "dam break", just consider modifying this in the manuscript. | Thanks for your advice, I modified this in the manuscript. |
| 3 | L15-L17. Improve redaction. | We rewrite the whole sentence as follows: "This study presents a comprehensive analysis of the characteristics of two hazards and the resulting damage to buildings from the cascading hazards". **(see Page 1 and L. 15-16)** |
| 4 | L70-L71. Check grammar | We rewrite the whole sentence as follows: "Our field investigations have revealed that the pattern of damage to buildings in the confluence area of debris flow and flood is distinct from those observed in areas affected by debris flow alone or by flood alone." **(see Page 3 and L. 70-71)** |
| 5 | Improve the quality of Figure 1. Maybe, ".eps" file type will be better to make the name of the province readable on the map. And make the font size bigger. Same for the legend. | We redraw this figure. |
| 6 | L116. The units must be better written as: "4700 m.a.s.l.". Check this in the whole manuscript. | We corrected the error in the whole manuscript. |
| 7 | L117. Avoid using qualifier words, e.g., "hot", "humid", or | We rewrite the whole sentence as follows: |

| | | |
|---|---|---|
| | "abundant rainfall". Instead, write the numbers, such as "minimum average temperature", "maximum average temperature", and so on. | "The average annual temperature is 16.2 ° and the average annual rainfall is 949 mm." **(see Page 4 and L. 115-117)** |
| **8** | L127-L139. The calculation of the slopes is wrong, I think. Correct this in the whole document. Thinking of this, it is just a typing mistake or were they introduced in Manning's equation in this way? Please, be completely sure about this. | The slope is calculated based on the channel profile; I think it's right. I modified the slope value as decimals **(see Page 5 and L. 126-135)** |
| **9** | L131-L132. This is not truly saying something. Consider being very specific about what you describe here or remove the sentence. | We rewrite the whole sentence as follows: "The field investigation indicates that debris flow initiated in the area above elevation of 1990 m a.s.l." **(see Page 5 and L. 128-1129)** |
| **10** | L132-L133. Improve redaction, e.g., the "valley" word does not seem to be suitable for this description with slopes of about "600%", if it is correct. | We use "channel" instead the "valley". |
| **11** | Improve the figure quality, to see better the date and time. | We redraw this figure. |
| **12** | L160-L162. Improve the redaction of the caption. | We rewrite this as follows: **(see Page 10 and L. 219-221)** "Figure 3 Illustration of the hazard cascade process: (a) the normal flow of river flow before the occurrence of debris flow; (b) debris flow blocks the river, creating a dammed lake that destroys the railway, roads, and buildings; (c) the dammed lake bursts, causing a flood that damaged and the road and buildings." |
| **13** | L172. Correct to "Digital Elevation Models". | We correct this mistake. |
| **14** | Table 1. Correct "Rn". | We correct this. |
| **15** | Why use the Manning equation and HEC-RAS to model this type of natural process? Because there are many different models and codes might help to obtain better results as they account for more variables and parameters. | Thanks for your advice. We first use many empirical models based on characteristics of debris flow dam and barrier lake to calculate the peak discharge of the dam break flood, the results vary widely. Considering the uncertainty of the characteristics of the debris flow dam and barrier lake, we used the Manning equation to calculate the peak discharge of dam-break flood. The |

| | | HEC-RAS is often used to model the flood process, See the following articles: Butt, M. J., Umar, M., & Qamar, R. (2013). Landslide dam and subsequent dam-break flood estimation using hec-ras model in northern Pakistan. Natural Hazards, 65(1), 241-254. Mozumder, C., Tripathi, N. K., & Tipdecho, T. (2014). Ecosystem evaluation (1989–2012) of ramsar wetland deepor beel using satellite-derived indices. Environmental Monitoring & Assessment, 186(11), 7909-27. |
|---|---|---|
| **16** | Better support how did you select or compute these Manning coefficient values. Be very specific to make the methodology replicable and/or applicable to other circumstances. | The Manning coefficient values were determined based on the suggested values in the HEC-RAS 5.0 Reference Manual. We add the reference **(see Page 7 and L.183)** |
| **17** | L217. Check the original paper out for this equation. Be very specific about what this equation describes, which is the "average" total pressure | This equation is calculated as the average total pressure, we rewrite it as follows: **(see Page 7 and L.190-192)** "Hazard intensity parameters were applied, such as flow depth and average total impact pressure, with average total impact pressure calculated as $P=\rho v^2+0.5\rho gh$ (Zanchetta et al., 2014) where P is the average total impact pressure, ρ is the flow density, v is the velocity, and h is the flow depth" |
| **18** | L258. Write the percentage of the "relative error" between the two calculations. | We rewrite the whole sentence as follows: **(see Page 12 and L.258-260)** "resulting in a flow discharge of 2273 m3/s with a relative error of 18% which is comparable to the result obtained by Manning's equation." |
| **19** | Improve the quality of figures 11, 12, 13 and 14, and increase the font size. | We rewrite the whole sentence as follows: **(see Page 20 and L.409-413)** "The impact pressure thresholds for Zones II and III, where vulnerability is equal to 1, are 75 kPa and 110 kPa, respectively. For the same impact pressure and inundation depth, the damage to buildings in Zone (II) is greater than that in Zone (III). " |
| **20** | Improve the quality of figures 11, 12, 13 and 14, and increase the font size. | We redraw this figure. |

| 21 | Discussion has to be improved by emphasizing the benefits and disadvantages of using this methodology to analyse buildings' structural vulnerability, based on debris flows and floods. | We added this discussion as follows: **(see Page 25 and L.510-521)**

"This study presents a comprehensive analysis of the damage to buildings resulting from a large-scale debris flow and outburst flood hazard cascade. The study develops building vulnerability in different areas of the confluence zone, which is useful for building risk assessment and management along the riverbank. However, some uncertainties and limitations are involved in vulnerability analysis. Firstly, the study did not consider the building's physical characteristics, such as shape, orientation, and maintenance condition. Secondly, in the area affected by the two hazards, the capacity of buildings first damaged by debris flow had declined, leading to a higher failure probability under the impact of sequential flood (Luo et al., 2020). The study analyzed the buildings' structural vulnerability based on debris flows and dam-break flood separately, and did not consider the building response to the primary debris flow or quantify the cumulative effect of the debris flow and the dam-break flood (Luo et al., 2023). A physics-based vulnerability model is required to quantify the dynamic evolution of building vulnerability."
In addition, we supplement the results of the debris flow simulation and adjust the structure of the paper. |

---

## Referee Report (RR1)

**Review of: The vulnerability of buildings to a large-scale debris flow and outburst flood hazard cascade that occurred on 30 August 2020 in Ganluo, Southwest China**

Overview:

The manuscript describes a hazard cascade where a debris flow dammed a river, causing upstream flooding and a downstream outbreak flood. Separate hydraulic modelling analyses were completed for the debris flow and subsequent flooding and outbreak flood. Three zones were defined, with primary debris flow impacts, secondary flooding/outbreak flood impacts, and sequential debris flow, then flood impacts, and vulnerability curves were defined. The model results were used to assess the contribution of the different components of the hazard cascade to the expected levels of building damage.

General comments:

1. The number of significant figures used throughout the manuscript should be representative of the level of confidence in the numbers reported. For example, discharge estimates coming from empirical relationships are reported to four significant figures, which suggests a level of precision that is not appropriate for these relationships, and the volume of the landslide dammed lake is reported to six significant figures.

2. The introduction and conclusion focus on confluence zones between debris-flow prone catchments and larger rivers. Would the approach developed in this work apply more generally to any area downstream of a landslide dam? In the introduction "confluence zone" should be defined, how far upstream/downstream is considered in this zone?

3. I would recommend the authors more explicitly compare the results of their simulations with the field observations. There is description of the actual event, and of the model results, but it's not clear as written where the model performed well or did not. This is important for evaluating the overall approach for representing the hazard cascade described in this manuscript.

Specific comments:

Line 29: Would "occasionally" be more appropriate than "frequently"? In my experience landslide dams formed by debris flow deposits are not common.

Line 32: This could be a good point to introduce how you are defining "confluence zone" in this work (see general comment 2).

Line 42: If the six steps of the Argyroudis et al. (2019) methodology are relevant to this work, they should be listed, otherwise you can delete "is comprised of six steps and".

Line 50: 300% increase in damage relative to what?

Lines 103 – 105: Can this geological description be better integrated into the rest of the work? Are there specific geologic units which tend to have more landslide events, etc.?

Lines 107 – 109: Provide a reference for the earthquake record.

Lines 118 – 120: Provide a reference for the climatic data.

Lines 126 – 127: The final sentence of this paragraph can be deleted because the information is presented in the following paragraph.

Lines 148 – 159: Provide a reference for the photogrammetry software/analysis technique used?

Table 1: Debris flow density – have the authors assessed how similar the debris flow at this site is to the debris flows that were used to generate the density relationship from Hu et al. (2019)? The fact this relationship uses a seventh order polynomial suggests the relationship is likely very highly fitted to the data. It should also explicitly state what the clay fraction is calculated by (I presume by mass).

Debris flow peak discharge – were the results from the equation cross checked against other methods (for example, volume-discharge relationships, see Ikeda et al., 2019)? The H term in the $n_c$ calculation is not defined in the description column.

Line 161: Provide references for using FLO-2D for debris flow simulations.

Table 2: How were the model parameters used selected? I would recommend changing the table title to "FLO-2D model parameters used in the debris flow simulation".

Line 167: The limitation of using the post-event DEM should be discussed at some point in the manuscript. It makes sense to use it, as it is information that there is higher confidence in, however it should be recognized that as the landslide dam breached the channel geometry would have been changing significantly during that time. Lines 176 to 177 state that "it was assumed that the peak discharge of the dam-burst flood formed the post-event terrain, which was adopted to simulate the dam-burst flood". The flood may have had its maximum erosive power at the time of the peak discharge, but the post-event terrain would have been formed over the entirety of the breach process.

Lines 167 – 168: On a related note, the choice of a one-dimensional steady state simulation should also be justified. A dam breach is a very transient phenomenon, could assuming a steady state at the peak discharge have any effect on the results for the end purpose of estimating damage levels? Would a 2D model be more appropriate for representing flows given the complexity of the topography?

Table 3: Similar to Table 2, the title should be "Parameters used in the flood simulation".

Figure 2: Add a description of what the red star indicates either on the plot or in the caption.

Lines 223 – 224: Can the authors provide some qualitative description of the deposit to go with the grain size distribution, i.e., clast supported, matrix supported, etc.? It states that material smaller than 100 mm was sampled, what was the maximum particle size observed in the field? Since the density was calculated based on the clay fraction, if there was a significant portion of the material larger than 100 mm that could influence the density estimation.

Lines 238 – 240: There should be some commentary on how well the simulated debris flow matches the observed event.

Lines 240 – 243: This information is not relevant to the simulation, and was stated previously, it should be deleted.

Line 250: It states that "empirical formulas" were used to verify the Manning's result, but only one formula is discussed following.

Line 258 – 259: There are more recent empirical relationships for landslide dam peak discharge (e.g., Froelich, 2022), I would recommend evaluating multiple empirical relationships given the importance of the peak discharge on the simulated flow depths and velocities.

Line 260: Please clarify, the relative error is relative to the result from the Manning's equation?

Lines 272 – 275: This information belongs in the methodology section.

Lines 275 – 276: Related to the comment on lines 223 – 224, how does the calculated maximum transported sediment size compare to the field observations of mobilized material?

Line 315: Please check that units are consistent throughout, $km^2$ is used here while $m^2$ is used elsewhere for area. See also lines 390 – 395 where pressure units switch from kPa to Pa.

Lines 346 – 348: This commentary seems out of place, it seems to be a more general statement on the effects of different hazards and local topography, not related to the event that is the focus of the manuscript.

Lines 361 – 362: A statement is made that buildings closer to the channel are always more vulnerable. This may not be true in all cases (for examples, buildings near a channel may be constructed to be extremely resistant to damage, and in that case the maximum vulnerability could be in areas with less inundation, lower impact pressures, but less robust construction. This statement should be changed to discuss this specific event, and not making unsupported generalizations.

Line 396: It is stated that there is no evidence the debris flow abraded the structure – can the authors connect this to their simulation results, are the simulations consistent with the field observations?

Line 431: Aside from being professionally constructed, are there any details on the construction of the buildings that lead to their greater resistance to damage? What makes them "tougher"?

Lines 469 – 472: It is unclear whether this is meant to be a general discussion on landslide dam outbreak floods, or if this is referring to this specific case. Please revise to make whichever you intended clearer.

Line 473: Should $110^5$ be $10^5$?

Line 483: Be specific as to how an "ordinary flood" is defined in this context. Similarly, an explicit comparison reference is needed for lines 535 – 537.

References:

Froelich, D.C. (2022). Peak discharge from a landslide dam outburst. Natural Hazards Review 23(2). https://doi.org/10.1061/(ASCE)NH.1527-6996.0000545

Ikeda, A., Mizuyama, T., Itoh, T. (2019). Study of prediction methods of debris-flow peak discharge. http://dx.doi.org/10.25676/11124/173152

---

## Referee Report (RR2)

General comments:

The authors of the study titled "The vulnerability of buildings to a large-scale debris flow and outburst flood hazard chain that occurred on 30 August 2020 in Ganluo, Southwest China " present a comprehensive analysis on damage resulting from a debris flows and dam break event. Their study presents a systematic investigation using modeling of the debris flow and outbreak event, analysis of damage patterns, and a livid discussion of the ensuing damage from the hazard chain. Their results are very interesting and make a meaningful contribution to the growing field of multi-risk management and mitigation.

My recommendations for this manuscript mainly revolve around improvements to writing and presentation of the information in this study. Though the results are very detailed and methodology clearly described, the presentation of results and the writing may require editing, as well as additional information and figures to support some statements. In the results section, the model and description of damage caused by the flood are combined. Furthermore, the results are missing information on validation and verification from their HEC-RAS simulations to give readers an idea of the performance of the model. There are segments in the results that describe the model performance, but an overview of the performance is not clear in this current version. Furthermore, the presentation in this paper could be improved by creating a segment with figures dedicated to describing and presenting the damage to buildings before discussing the model results. I therefore recommend minor revisions that the authors may want to consider for their manuscript, before publication.

Specific Comments:

L55-67: The authors could further expound on how they arrive at "..this approach may not apply directly to debris flow-dam-burst flood hazard cascade." Although the literature preceding establishes the importance of their approach, L65-67 seems to be missing a connection.
L68-69: This statement could be improved with support from a reference.
L86-87: It may be more appropriate to refer to Fig 1 here.
L142-143: Information on the period of time before and after the disaster these drone photos were taken would be nice to state.
L152-153: Does this error represent both vertical and horizontal accuracy?
Table 1: Description parameters are clear, although some lines may require the reference ( i.e. the method for calculating $nc$ was deduced from analysis of viscous debris flows in Huoshao gully in China.). It was unclear to me if some information were statements from the reference under the Category of Calculation column.
L199: This line may need a return period analysis or a reference to support this claim.
L210: 1,050,000 could be abbreviated to 1.05 M to improve readability
Figure 2: X-axis dates and times are difficult to read, authors may want to consider improving this figure by nesting times within dates on the axis. Furthermore, the star indicating the date and time of the event is not indicated in the caption, and could be made clearer visually with a larger or more emphasized icon.
Figure 5: Y-axis title has a typographical error.
L235-236: Unclear where the buildings are. In neither figures prior (Fig. 4 or Fig. 1) are the buildings depicted. Although Fig. 3 illustrates this, it would be appropriate to either add another panel to Fig. 6 showing the buildings, or to locate them already in Fig. 4.
Figure 6. Insert captions a and b area uncles; legend is not readable (too small) and the figure caption does not indicate which figure corresponds to which.

L247: The authors should consider supporting this claim with a time series from their simulations. This statement is missing a supporting figure or data. Until this point, they only present the spatial aspect of the flood event (Fig. 6.).

L247-250: These lines may be better in the methodology section rather than in the results section.

L251-253: How do these observations line up with the model results? A location indicator in Fig. 6 may improve the presentation and connection for this statement.

L256-257: The estimated volume of the barrier lake could be presented in this line.

L260-262: It is unclear what two scenarios are being introduced in this line. This may be a good point to begin a new paragraph, and clearly indicate what scenarios are being introduced in Fig. 7.

L262-264: This is a very important statement, but may not be appropriate here. Authors may want to consider separating the paragraph that locates the flood-damaged areas with an improved figure so that readers can spatially locate the flooded areas and damaged buildings with the model results.

L267: It is unclear where this scenario was formally introduced.

L267-279: It is unclear from the presentation of this statement if this was observed by the authors during their field investigation or is this based on literature.

L283-284: This statement could be combined in a separate paragraph describing the damage, and would be better appreciated by readers if located or indicated in any of the figures.

Figure 7. Legend size could be increased to improve readability.

Figure 8. Caption could be improved to indicate that sections here refer to sections in an earlier Figure.

Section 4.4: The authors may want to consider reporting this section and description of the damages earlier in the results. This would greatly enhance readability and garner appreciation for the magnitude of the event before reading into the model results.

L302-310: The authors may want to refer to Fig. 3 when describing this process.

L'308-310: Connecting this statement to the time series in Fig. 2 could help readers appreciate this description.

L316-317: This would be better in a summary section of the damage. It is difficult to follow the presentation of the event.

Figure 9. This figure could be presented earlier to improve readability. The current order of the sections make it difficult to follow.

L346-348: It is unclear if statements on the different damage zones are not static because they were determined by the damage observed or the model results. The authors may want to further expound or clarify this detail.

L380-381: This statement requires reference to a figure.

Figure 11. Building labels are difficult to identify, authors may want to consider enhancing the text color to improve identification and readability.

L384-392: Building references require references to Figure 11.

L393-394: Building 26 requires reference to Figure 11.

Figure 12 and Figure 13: Figures could be enlarged, and presented separately instead of in two columns to improve readability.

L427-428: The authors may want to expound, reference or describe the different damage patterns to buildings in their study that leads to this disparity.

L428-432: This may be more appropriate in the discussion section.

Figure 14 and Figure 15: Figures could be enlarged, and presented separately instead of in two columns to improve readability.

L444-449: May consider rewriting these sentences, as they run over each other and are difficult to read. It is also unclear how the supporting references support the final statement in its current state.

L453-466: This paragraph may be more appropriate in a limitations section in the discussion, rather than in the results.

L465-466: The authors could consider describing the detailed data required from the insights in this study to direct future research and add outlook from their work and experience.

L469-478: References to appropriate figures from earlier sections may greatly improve readability and aid readers in understanding where statements in this summary paragraph are derived from.

Figure 17. The authors may want to consider moving this figure to the results section or creating a section to show this. This is an important result, and may require information on the model verification and validation results supporting the areas delineated in this map.

L510-521: The authors may want to consider starting the section with this paragraph.

L520-521: Unclear if this is a conclusion from this study or a statement from a reference or literature review.

L514-515: The authors raise an important point, but could enhance their discussion by adding insight to how these physical characteristics can affect vulnerability.

L540-541: Unclear if this is applicable to only this study site or if this a general application to other study areas, based on the results and methodology presented.

L549-544: This is an important conclusion. Further support in the discussion section would aid readers, give outlook, and inform future research directions.

---

## Author Response (AR2)

Dear Editors and Reviewers:

On behalf of my co-authors, we thank you very much for giving us an opportunity to revise our manuscript, we appreciate editor and reviewers very much for their positive and constructive comments and suggestions on our manuscript. We have studied the comments from editors, and we have tried our best to revise our manuscript according to the comments.

We have made the following major changes in this article:

First, the article was revised and supplemented based on the editor's suggestions. Second, minor errors in the text and figures were corrected without affecting the conclusions. Finally, minor grammatical errors were also addressed.Please review the manuscript with revision marks.

In addition, A clear PDF version of the images in the article has been uploaded as an attachment in the system.

The main corrections in the paper and the responds to the editor's comments are as flowing.

| No. | Comments | Response |
|---|---|---|
| 1 | Providing more information about the model validation and performance, and at improving the section dedicated to the results | A comparison between the debris flow and dam-burst simulation results and the actual survey results has been added. For the debris flow, the simulated deposition area was compared with the actual deposition area. Additionally, two buildings were selected to compare the simulated maximum flow depth and the actual deposition height as follow**(see Figure 6, Figure 7, see P 11 and L. 238-243)** *"The deposition area obtained from the simulation is 0.15 km², which is close to the area measured from the UAV image, approximately 0.16 km². The thickness of the sediment deposits ranged from 5 m to 15 m, with an average value of 7 m. Fig. 7 shows that the debris flow buried one floor of Building 3 and nearly two floors of Building 4 (locations indicated in Fig. 6). The simulated maximum depths at Buildings 3 and 4 are 3.2 m and 5.5 m, respectively, close to the actual deposition heights.."* |

| | | For dam-burst flood, the simulated flood area was compared with the actual inundation area. Additionally, two buildings were selected to compare the simulated maximum flow depth and the actual inundation height as follow(**see Figure 8, Figure 9, see P 12 and L. 267-274**)

   *"The simulated inundation area of the outburst floor is 0.18 km², which is consistent with the field investigation result with an error of 1.1%. The flood completely submerged all buildings on the left bank near the middle of the river channel, and the buildings on the river terrace on the right bank were strongly eroded. The maximum water depth and velocity of the dam-burst flood were 13.96 m and 8.24 m/s, respectively, which were 1.24 and 1.31 times higher than those of the ordinary flood, respectively. The maximum depth of the dam-burst flood at locations of Buildings 8 and 26 were 6.4 m and 3.7 m, respectively (Fig. 9) (building locations indicated in Fig. 8), which are close to the result obtained by field investigation."* |
|---|---|---|
| **2** | Before presenting the model results, a description of the damage from the event should be included. | First, an overview of the damage was added in the Instroduce section as follow (**see P3 and L. 78-82)**
   *"On August 30, 2020, a catastrophic debris flow and dam-burst flood occurred in the Niri River, Ganluo County, Sichuan Province, Southwest China. The debris flow-flash flood event killed 3 people and caused serious damages to local infrastructure, including the destruction of 110 buildings, the Chengdu-Kunming railway bridge near the gully mouth, 1.2 km national road, and 5 highway bridges along the main river."*

Then, a detailed description of the damage was added in the hazard cascade section as follow(**see P8 and L. 213-218)**
   *"The debris flow swept away the railway bridge that crossed the gully mouth and impacted the national road across the river. It also destroyed the buildings close to the gully mouth and those on the opposite bank of the main river. Approximately 40 minutes later, the debris flow dam was breached, triggering a high-magnitude flash flood that damaged the national road and buildings near the altered flooding path (Fig.3)."* |

---

## Author Response (AR3)

Dear Editors:

On behalf of my co-authors, we appreciate editor and reviewers very much for their positive and constructive comments and suggestions on our manuscript.

We have made the following changes in this article:

Satellite image in the Figures 6, 8, 11 and 19 was obtained form the website (https://www.jl1mall.com/), the information was added in the figure captain.

The other maps was entirely created by us.